# The Effect of Auriculotherapy on Situational Anxiety Trigged by Examinations: A Randomized Pilot Trial

**DOI:** 10.3390/healthcare10101816

**Published:** 2022-09-21

**Authors:** Andreia Vieira, Paula Sousa, Alexandra Moura, Lara Lopes, Cristiane Silva, Nicola Robinson, Jorge Machado, António Moreira

**Affiliations:** 1ICBAS, Institute of Biomedical Sciences, University of Porto, 4099-002 Porto, Portugal; 2CBSin, Center of BioSciences in Integrative Health, 4250-105 Porto, Portugal; 3ESSSM, Superior Health School of Santa Maria, 4049-024 Porto, Portugal; 4LSBA, Institute of Health and Social Care, London South Bank University, London SE1 0AA, UK; 5ESDRM, Sport Sciences School of Rio Maior, 2040-413 Rio Maior, Portugal

**Keywords:** auriculotherapy, anxiety, acupressure, auricular points, salivary cortisol, randomized controlled trial, parasympathetic nervous system, brain modulation

## Abstract

Background: Auriculotherapy may activate the parasympathetic nerve system and reduce anxiety levels. Short-term auriculotherapy’s effects and safety on university students’ anxiety levels was assessed prior to exams. Methods: A randomized, controlled pilot trial was conducted. The day before the exam, university students were randomly allocated to the auriculotherapy group (AA, *n* = 13) or the waiting-list group (WG, *n* = 13). Baseline measures were taken 4 weeks before the exam at Time point (TP 0); at 7.30 a.m. on the day before the exam (TP I); at 11 a.m. before auriculotherapy (TP II); 30 min after AA (TP III); and at 7.30 a.m. before the exam (TP IV). The outcomes were the State-Trait-Anxiety Inventory (STAI); quality of night-sleep, Visual Analogue scale (VAS) for anxiety, and salivary cortisol. Adverse events were also recorded. Results: A total of 26 students participated in this study and became more anxious as assessed by STAI in TPII (*p* = 0.002) and TPIV (*p* = 0.000) than TP0. AA reduced the STAI in TPIII (*p* = 0.045) and PIV (*p* = 0.001) and the VAS (*p* = 0.012) in TPIV. Cortisol was reduced in TPIII (*p* = 0.004), and the AA slept better (*p* = 0.014) at TPIV. Discomfort at the auricular site was reported in only one AA participant. Conclusions: Auriculotherapy appeared safe and effective in reducing anxiety levels before university exams.

## 1. Introduction

Anxiety is one of the most natural body responses, and it is fundamental for humans to a given threat [1,2]. However, anxiety can become pathological due to hormonal imbalance during physical and emotional challenges over time [3].

Mental distress has been identified in undergraduates [4]. Young students are the most predisposed to stressful life events, especially those pursuing higher professional education in a competitive setting [5]. A high prevalence of anxiety among health care professional students has been identified globally [5,6,7]. There are various factors that are responsible for the decline in the mental health of students with healthcare degrees, including academic pressure [8], an increased workload [6], financial issues [8], sleep deprivation [9,10], and being exposed to patient death [7]. 

Auriculotherapy (AA) has been employed for approximately 2500 years in Chinese civilization [11]. It is a technique used to diagnose and treat physical and psychosomatic dysfunctions by stimulating a specific point in the ear [12] using needles [13], seeds [14], magnetic stones and lasers [15], bloodletting, moxibustion, electric stimulations, or massaging the auricular points [16]. In Europe, AA has been applied systematically and comprehensively since Doctor Nogier introduced the map of the inverted fetus in 1957 [17], and the idea that the whole body (e.g., the somatotopic/holography rule of points) is reflected by the ear [18]. The somatotopic perspective has received scientific support from double-blind studies investigating auricular diagnosis and treatment for several disorders [19,20,21,22,23,24]. The current understanding of the mechanisms underpinning auriculotherapy is based on the embryological hypothesis [18,25] and the innervation of the ear [25]. It is recognized that the AA technique might work in situational anxiety because the puncture of a reflex point in the ear elicits responses of the reticular formation and through the sympathetic and parasympathetic nervous systems activation [22,26].

In recent randomized crossover trials [27], AA with needle intervention reduced pre-exam anxiety and increased sleep duration in medical students, where the intervention was superior to a placebo procedure in treating anxiety. It seems that AA can have a critical role in treating situational anxiety. However, AA’s effectiveness is subject to the acupuncturist´s professional qualification and assertiveness with which type of intervention and points are selected in their therapy.

In the past, we have shown a tendency to reduce anxiety levels after 30 min with AA, although the effect was significant after 48 h from the experimental session [28,29]. However, these results were related to AA being performed 2 weeks before students’ examinations, so we designed a new study with new variables and outcomes. The question that guided this study was: Is AA effective and safe on anxiety levels provided the day before an exam compared to a waiting-list group (WG)? To address this question, anxiety levels were measured using STAI as the primary outcome, with the quality of night sleep, the VAS for anxiety, and levels of salivary cortisol as the secondary outcomes. Therefore, we expected a reduction in anxiety in the AA group but not in the WG. Our main aim was to assess the effectiveness of AA in reducing anxiety in students according to the STAI and report its safety. As secondary goals, we (i) evaluated AA effectiveness in decreasing secondary outcomes; and (ii) investigated which auricular side of the ear was more reactive with appropriate equipment before AA application.

## 2. Materials and Methods

### 2.1. Study Design

An experimental, prospective, randomized, controlled, and single-blinded pilot study was conducted in December 2021 at the Private Superior Health School in Porto, Portugal. The Institutional Ethics Committee approved the research project on the 3 July 2021 (reference no. 2021-11). Before enrolling the participants, the authors registered the protocol at clinicaltrials.gov (registration number NCT05042778). 

### 2.2. Participants and Setting 

All participants agreed to provide written informed consent following the Declaration of Helsinki guidance and were made aware that they could withdraw from the study at any time. According to the eligibility criteria, the participants were approached by one of the authors (CS), who informed them about the study before a routine lecture 10 weeks before their exam. The students interested in participating in this research provided their email addresses, and the information sheet was sent in September 2021. They were not given any incentives to engage in this trial. One month prior to the examination at the university, all students interested in taking part were accepted for the screening visit. The screening started by collecting their signed informed consent, completing questionnaires, such as demographic data, the trait and state anxiety questionnaire (STAI) and the inventory psychopathological symptom questionnaire [30] (Figure 1).

For inclusion in our study, the students were recruited if the following inclusion and exclusion criteria were met: 

Inclusion Criteria:Able to sign the informed consent.Aged above or equal to 18 years or older.Unfamiliar with AA.

Exclusion criteria:Psychological disorders based on Brief Symptom Inventory scale.Students with any neurological disease, cardiovascular disease, renal disease, or chronic disease history.Known to be pregnant.Participants taking psychiatric medication.

The baseline study visit was conducted the day before the exam. It should be noted that before the formation of two groups and to ensure the highest homogeneity, the students screened were previously stratified into four groups according to their trait anxiety level as: first group were students with STAI from 20 to 35 points (6 students); the second group were subjects with STAI from 36 to 50 points (12 students); third group were students with STAI from 51 to 65 points (8 students); and the fourth group were subjects with STAI from 66 to 80 points (0 students). Subsequently, each stratified group was randomly allocated into two groups: (1) WG (participants who did not receive the intervention) and the AA group (participants who received AA 1 day before their exam). Thus, each student extracted at random out of a black bag a slip of paper that had the initials A or W. Each initial corresponded to: A = AA group, WG = no intervention. After the group allocation, the first investigator informed the acupuncturist of the participant’s assignment before any intervention.

### 2.3. Study Intervention

A licensed acupuncturist (holder of civil liability insurance) with more than ten years of experience applied in the auricular area of the AA group, two steel balls (Ø 1.2 mm) covered by a transparent anti-allergic adhesive (Ø 7 mm). The points selected were F6 and J13 (Figure 2) based on the International Auricular Nomenclature guidance [31].

For the intervention group, we chose frequently used areas from most of the included studies cited in our recent systematic review [27] and from the author’s clinical experience. Both auricular points were chosen only from one ear (right or left ear) based on an assessment of the reactive site (e.g., painful or sensible by touching with appropriate material). The procedure took approximately 5 min, and the steel balls were left in the same place until the end of the exam. The students were instructed to manually manipulate the spheres during the day.

All WG participants were given the opportunity to receive the same treatment as the AA group after the exam at the end of this trial.

### 2.4. Types of Outcome Measures

Outcomes were measured at different times (Figure 3), where the outcome’s assessor was unaware of the participants’ allocation to the study group as a third party performed it. The screening visit occurred one month before the exam (TP 0) to assess whether participants met the inclusion criteria. The baseline visit started on the same day as the screening day (4 weeks before the exam), collecting STAI (state anxiety form) at TP 0. The TP I was at approximately 7.30 a.m. 1 day before the exam, and the TP II corresponded to before the AA was administered. The TP III was performed 30 min after AA, and the TP IV was completed at approximately 7.30 a.m. before the exam.

### 2.5. Primary Outcome

The anxiety levels were measured using the Portuguese version [32] of Spielberger’s STAI form Y1, ranging from 20 (low anxiety) to 80 (highest level of anxiety). The score varies from 20 to 80 points, where 20 to 35 points mean not anxious, 36 to 50 points is considered a little anxious; 51 to 65 points means moderately anxious; and finally, 66 to 80 points means the participant is considered very anxious. This STAI form Y1 scale assesses the state of anxiety (situational anxiety), while the STAI-Y2 form assesses the trait anxiety (subjects’ anxiety personality tendency). The STAI form Y2 was used only in TP 0, while the difference in STAI form Y1 for anxiety was compared during TPs 0, II, III, and IV.

### 2.6. Secondary Outcomes

The difference in VAS for anxiety was used during TPs II, III, and IV. It consists of a horizontal or vertical line, 100 mm long, which has marked the classification “totally calm and relaxed” at one end and at the other the category “Worst fear imaginable”. The respondent should mark the point representing the degree of intensity of his anxiety. The scale is reliable and correlated with STAI-Y1 (*p* < 0.0001) for the level of anxiety [33].

Saliva samples were taken at TPs I, II, III, and IV for the AA group and at TPs I, II, and IV for WG. The salivary collection was performed in a specific device called “salivette”. Both groups collected the samples at approximately 7:30 a.m. fasted before and 30 min after the AA. The saliva was extracted from the salivette cotton by centrifugation, and the cortisol measurement in saliva was performed through immunoassay by electrochemiluminescence, according to the manufacturer’s instructions (range 0.005–3 μg/dl, IBL International Cortisol Saliva ELISA, RE52611, Hamburg, Germany). The saliva samples collection was followed by Amorim, D (2022) research accepted protocol [34].

We considered an adverse event “Any untoward medical occurrence in a patient or clinical investigation subject administered a medicinal product and did not necessarily have a causal relationship with this treatment”. The acupuncturist documented all adverse events since the AA until Time point IV by using a questionnaire asking about the symptom, incidence, hour, and date of the event and if the subject could describe from zero to ten values the severity of the symptom. 

On the morning of the exam (TP IV), the participants were asked to classify the quality of sleep the night before as (1) worse, (2) no change, or (3) better than the quality of sleep during the previous week.

### 2.7. Statistical Analyses

The pilot sample size was calculated based on the confidence level of 95%, margin of error 5%, population proportion of 50%, and population size of 26. So, the minimum total sample size of 25 participants needed to have a confidence of 95% that the real value was within ±5% of the measured value. 

Data analysis was performed using IBM SPSS Statistics Software for Windows (Version 28.0.1.1, IBM Corp., New York, NY, USA). Normality of distribution was evaluated in the groups before the analyses assessed by the Shapiro–Wilk test. Parametric analyses were carried out in cases where normal distribution assumptions were met, and nonparametric tests were used in cases where normal distribution assumptions were not met. Continuous variables were presented using mean, standard deviation, median, minimum, and maximum values. A comparison of two independent groups was carried out with an independent sample *t*-test/Mann–Whitney U test. The McNemar test was used to analyze sleep quality the night before the anatomy exam. Missing values > 5% were inferred with multiple imputation analysis by the regression method. Statistical significance was considered unilateral at the level of *p* < 0.05.

## 3. Results

A total of 80 students were pre-screened, but only 30 were interested in participating in the research. Table 1 shows the demographic data through each stage of the study. In the end, a total of 26 students (*n* = 26, where 20 were women and six men, the age 20.2 ± 1.78, the weight 67.6 + 18 kg, the height 1.66 + 0.07 m, and the coffee consumption of 1 + 1.2 coffees per day) were recruited and completed the study.

The mean age difference between each study group did not reach statistical significance. In addition, there were no statistically significant differences in gender, height, weight, alcohol intake, smoking, and coffee habits between the two groups (*p* > 0.05) (see Table 1). Based on trait anxiety assessed using the STAI form Y2, 53.8% of participants were considered not anxious, 26.9% were judged as moderately anxious, and 19.2% showed little trace of anxiety. There were no significant differences between AA and WG regardless of demographic data analyzed by the independent samples *t*-test. All the participants were unfamiliar with acupuncture and with no psychiatric or thyroid disorders. Only one participant was excluded due to psychotropic medication (*n* = 1), and three did not attend. Regarding the reactive points, 88% of the participants reported the J13 point from the right side as more sensitive to touch than the left side of the ear, and 58% reported the F6 point from the left side as more sensitive than the right side.

### 3.1. Anxiety Levels One Month, 24 h, and Immediately before Exam

Analyzing the state anxiety levels based on paired-samples *t*-test, assessed with STAI, the participants were more anxious 24 h (Figure 4) before the auriculotherapy (M = −6.96, SD = 10.4; t(25) = −3.40 *p* = 0.002) and immediately before the examination compared to one month previous (M = −11.54, SD = 10.88; t(25) = −5.40 *p* = 0.000).

### 3.2. Anxiety Levels Using State-Trait-Anxiety-Inventory

Table 2 and Figure 5 shows the mean and standard deviation of the STAI anxiety score for each group. The state significantly decreased in the AA group compared with the WG after 30 min t(24) = 2.11 *p* = 0.045 and in the morning before examination t(24) = 3.96 *p* < 0.001.

### 3.3. Anxiety Levels Based on Visual Analogue Scale

Table 2 and Figure 6 reveal the mean and standard deviation of the VAS anxiety score for each group. The state anxiety levels significantly decreased in the AA group compared with the WG before examination U = 100,500; *p* = 0.016.

### 3.4. Salivary Cortisol

Table 3 and Figure 7, demonstrate a decrease in salivary cortisol after AA, although we did not find a significant difference between groups except immediately before AA where the salivary cortisol was significantly higher t(16) = −1.775, *p* = 0.047 in the AA group compared with WG.

Besides that, the salivary cortisol significantly decreased t(8) = 2.291 *p* = 0.026 immediately 30 min after AA in the AA group (Table 4).

To minimize the missing data and bias in the salivary cortisol sample (between 38.5%–46.2% of missing data), we performed multiple imputations by regression method (five imputations). Surprisingly, the Mann–Whitney test for independent samples corroborated the results obtained previously, therefore we decided to use the original data (Table 3 and Table 4).

### 3.5. Sleep Quality

AA group increased sleep quality compared to the WG confirmed by McNemar–Bowker, Test value = 10.667 *df* = 3, *p* = 0.014. As shown in Figure 8, the AA group increased sleep quality compared to the WG.

### 3.6. Adverse Events

Of the 13 participants in the AA group, only one adverse event was described by one participant. The student reported discomfort during the night caused by the pressure of the ear on the pillow when lying down on their side. The discomfort was rated as 4/10 through a VAS. This event was considered non-severe as the discomfort disappeared after removing the AA as previously instructed.

## 4. Discussion

This trial aimed to assess the clinical effect and safety after the application of one AA session provided to university students before their school examinations in order to reduce their anxiety levels.

The results show that the participants became more anxious 24 h before the exam and even more anxious immediately before the exam than one month earlier. According to STAI and VAS outcomes, AA reduced anxiety levels 30 min after the AA (also shown by salivary cortisol) and immediately before the examination, similar to several other studies [29,35,36,37,38]. However, during our previous trial where we assessed the effects of AA on anxiety levels in students as well, we did not find strong evidence for anxiety reduction after 30 min, maybe because we performed the AA 2 weeks before the examination period or the intervention selected was not following an individualized assessment [28,29].

Salivary cortisol as a biomarker of anxiety and the hypothalamic–pituitary–adrenocortical axis (HPA) role is a well-established approach in psychological research, dating back at least 20 years [39,40,41]. For this study, we had in mind the stability of measuring salivary cortisol as an outcome once the proportion of salivary cortisol to total cortisol (blood cortisol levels) was approximately 1–2% in the lower range, and 8–9% in the upper range [9,40]. Thus, the researcher should treat the salivary cortisol levels with caution since they will not behave linearly to serum levels in response to a challenge or under conditions that affect cortisol binding globulin levels, such as oral contraceptives [42] and menstrual cycle [43]. Several other factors could also affect the salivary cortisol response, including gender and smoking [42], age [44], exercise [45], and awakening time [46]. Although the groups in our study were identical, the sample population was mainly female; consequently, menstrual cycle and oral contraceptive use could have also influenced the cortisol levels. Other researchers have reported the same issue where changes in the cortisol awakening response varied during the treatment period [47]. In our study, we tried to minimize those biases by introducing salivary cortisol collection in all groups at the same TP (the participants were all instructed to take their samples first thing in the morning) and measuring it immediately before and after the intervention. Recently, Usichenko and Wenzel (2020) found that salivary cortisol activity was lower before the anatomy exam than in the evening before the exam after AA compared with the WG [35]. Our pilot study also found a significant decrease in the AA group’s salivary cortisol levels immediately 30 min after AA.

Interestingly, we also found that the salivary cortisol measured in the morning before the treatment in the AA group showed a fall in the morning before the exam compared with WG. However, the relevance of these results carries no statistical significance due to the sample size. Perhaps a more comprehensive sample would have been crucial to obtaining the statistically significant effects of the present investigation and, therefore, it would be necessary to increase the number of participants to increase the magnitude of our results. Curiously, we also noticed that immediately before the intervention, the salivary cortisol showed a trend toward a steeper rise in the AA group compared with WG. From the author’s point of view, this finding might suggest that AA group participants also became slightly more distressed than WG before AA, maybe because the participants saw AA stimulation as a challenge. Research evidence indicates that psychological variables, including other conditions, such as novelty and unpredictability, are associated with increased HPA activity and cortisol release [17,46]. Nevertheless, in many of these studies, a considerable percentage of subjects did not respond (i.e., did not show elevated cortisol levels) after exposure to a stressor, perhaps due to anticipatory effects [47]. The sensitivity of the HPA to a combination of different situations may lead to less valid cortisol assessments as a specific indicator of anxiety. The interpretation issues presented by non-responders or those who demonstrate inverted responses make the cortisol measurements difficult to interpret [47].

To support the results obtained in Chueh et al. (2018) and Usichenko Wenzel’s (2020) study, we also found AA improved sleep quality compared with the night before. Surprisingly, we could not find any research about the effects of AA on melatonin levels, so perhaps it would be interesting to investigate melatonin assays in future research.

Regarding adverse events related to AA, Correa et al. (2020) reported that AA with needles could cause bleeding and headaches [48] and local pain [49]. For Tan Jing et al. (2014), the most often adverse events for AA with seeds are local skin irritation, discomfort, mild tenderness or pain, and dizziness [50]. In our systematic review of AA for anxiety [27], from 13 included studies, most trials (11 studies) did not assess adverse effects properly [28,37,49,51,52,53,54,55,56,57,58], and two studies did not report any side effects [59,60] after AA. In this pilot study, we only found one participant reporting discomfort. However, the symptoms disappeared immediately after the auricular steel balls were removed as previously instructed to the participant.

In this research, 88% of the participants reported the J13 point from the right ear side as more sensitive/painful than the left side, and 58% stated the F6 point from the left side as more sensitive/painful than the right side. We decided to personalize the treatment based on patient needs as AA literature and acupuncture schools recommended. The rationale behind this type of AA assessment is that some auricular areas have lower levels of electrical skin resistance than involving tissue [61], and that electrodermal differences are linked to autonomic control of blood vessels rather than increased sweat gland activity [61]. Therefore, the heightened tenderness of reactive AA points may be clarified by the accumulation of toxic, subdermal substances. So, the activation of specific auricular points could lead to site-specific neural responses in distinct brain regions [61]. Recently, the stimulation of the vagus nerve via the tragus region (J13 point) and triangular fossa (F6 point) region [62] in the ear has been demonstrated by functional magnetic resonance studies [26,63,64]. These studies have shown that the intrinsic and extrinsic brain feedback from auricular stimulations is the best way to neuromodulate brain plasticity for therapeutic purposes [49,65]. It is noteworthy that both above-described auricular points have been shown to activate the left amygdala, anterior cingulate cortex, and cerebellum [33] and can instantly and effectively generate changes in the prefrontal auditory cortex and limbic cortices [43].

Interestingly, in 2001, Wang and Kain observed that the anxiolytic effect at the F6 point was not as profound as the “relaxation” point, also localized in the auricular triangular fossa. The previous authors also agree that in general AA practice, there is always more than one auricular point selected for the treatment of anxiety [64]. As we have shown in our results, only 58% of the participants have referred to the F6 point from the left ear as sensitive/ painful with pressure compared with the right side. This means that 48% of the participants have chosen the F6 from the right ear. Because there are still authors insisting on doing AA research with pre-prepared protocols [19] instead of first following the basic AA principles with the treatment based on Chinese Medicine Criteria diagnosis [47], perhaps we must instead establish experimental models in AA in anxiety and then select frequent auricular points used that might not be the most appropriate for all the patients. We agree with the study of Huang W., et al. (2012), where possible, pragmatic approaches used in acupuncture trials (in this case, AA trials) could lead to more successful evidence [47].

## 5. Limitations and Conclusions

There are several limitations to this study. Firstly, the samples used in the current study were small, so other tests would need much larger sample sizes to provide reliable results. Our observations exclusively reflected the acute changes in anxiety levels after one AA session. Similar studies about AA have included long-term effects analyses, which we will address in future studies.

Secondly, our study did not assess the WG salivary cortisol 30 min after TP II, as we assumed the salivary levels would not be different. Additionally, we faced more than 30% missing data from WG’s salivary cortisol samples. Because the WG was instructed to collect their samples from home and did not need to wait for the AA approach, it is understandable that those participants forgot to collect the samples. Therefore, we strongly recommend a more extensive study to confirm our observations. Nonetheless, we remember future considerations regarding cortisol concentrations that are different throughout the day [65], which also has variability in demographic participants [46].

Moreover, we recognize the lack of a placebo group; however, in our previous study [28], the placebo group did not produce significant changes in the students anxiety ‘levels (either 30 min or 48 h after the auriculotherapy). There is a persistent controversy over the results regarding the placebo evaluation and non-pharmacological therapies research. On the one hand, there seems to be a constant conflict between recognizing the potential therapeutic benefit of the placebo effect and the ethical questions surrounding its use [66]. On the other hand, some researchers are against using a placebo in AA trials [67] as the external ear is a tiny organ with more than 93 recognized active stimulation points [68]. Following Appleyard et al. (2014), the word sham in acupuncture research has become confused with the term placebo. The authors explain in their work that there have been no actual placebo-controlled trials in acupuncture because the sham-controlled trials compared different acupuncture concepts. From this point of view, as these acupuncture concepts have led to different outcomes [16], it seems that there is a physiological basis for acupuncture [67,69]. Consequently, the notion of a sham–placebo acupuncture remains an aspect of research that needs more rigorous evaluation [69].

As we have already mentioned, the participants reported the left F6 point and the right J13 points were more frequently reactive in this trial. So, it would be interesting to assess in future research if the conjugation of the points would affect the results (e.g., stimulation of right F6 and left J13 is more effective than stimulation of left F6 and right side J13) or if making both points at the same time (e.g., stimulation of bilateral points, F6 and J13) was more effective than using just two points.

Finally, AA applied to the auricular areas innervated mainly by the auricular branch of the vagal nerve seems to show a reduction in situational anxiety triggered by school examinations and improved the quality of sleep in students compared with the WG.

Potentially, AA with seeds can be used as a complementary or alternative treatment for anxiety for the period before school exams as it appears safe and effective, with minimal adverse effects, rare contraindications, and a theoretically reduced cost.

## Figures and Tables

**Figure 1 healthcare-10-01816-f001:**
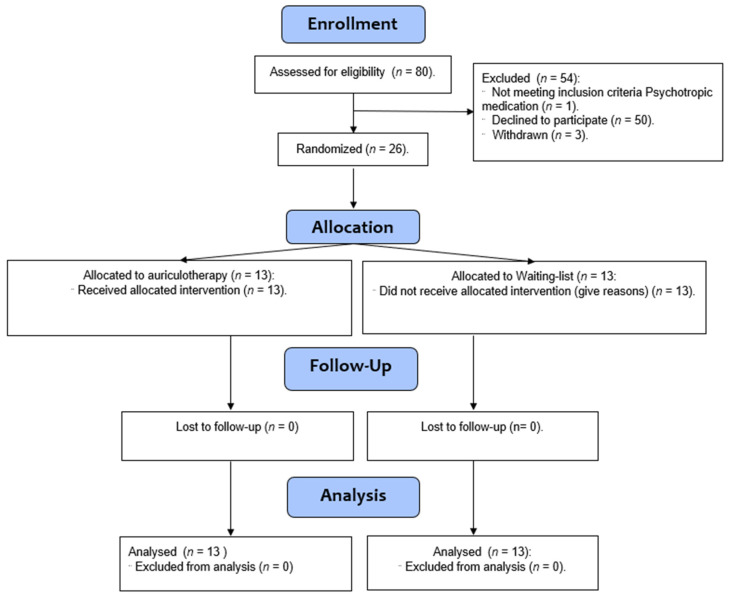
Study flow diagram based on Consolidated Standards of Reporting Trials (CONSORT) guidelines.

**Figure 2 healthcare-10-01816-f002:**
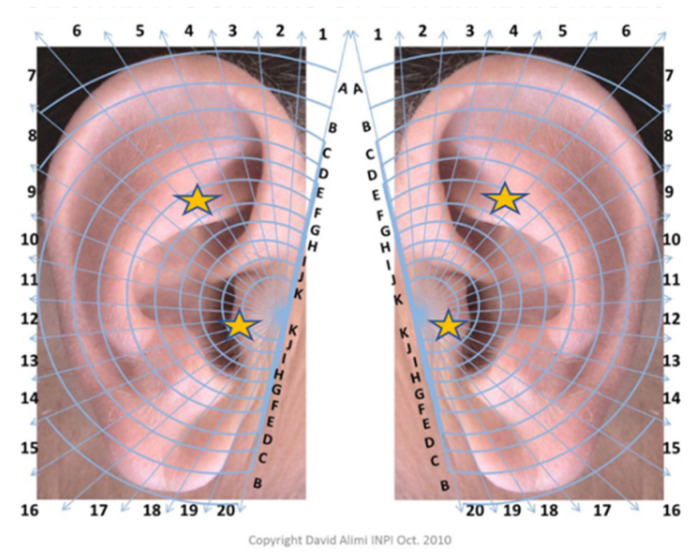
Segmentogram based on French Cartography University/scientific school of Paris (right and left medial auriculogram) from International Auricular Nomenclature. Legend: stars indicate the points applied to the auriculotherapy group (F6 and J13). The laterality chosen was based on the most symptomatic point (right vs. left ear). This cartography was used with permission [31].

**Figure 3 healthcare-10-01816-f003:**
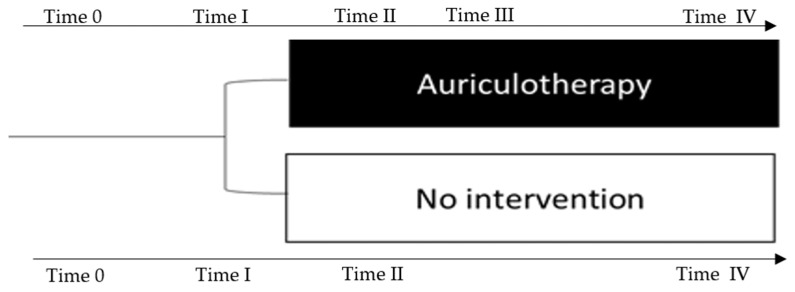
Timeline of the study session. TP 0: Screening and baseline period (one month before exam) outcome: State Anxiety inventory. TP I: morning (7.30 a.m.) 1 day before exam (outcome: sleep quality and salivary cortisol). TP II: Before (11 a.m.) AA (outcome: State Anxiety inventory, visual analogue scale, sleep quality, and salivary cortisol). TP III: 30 min after AA (outcome: State Anxiety inventory, visual analogue scale, and salivary cortisol). TP IV: morning (7.30 a.m.) before examination (outcome: State Anxiety inventory, visual analogue scale, sleep quality, and salivary cortisol).

**Figure 4 healthcare-10-01816-f004:**
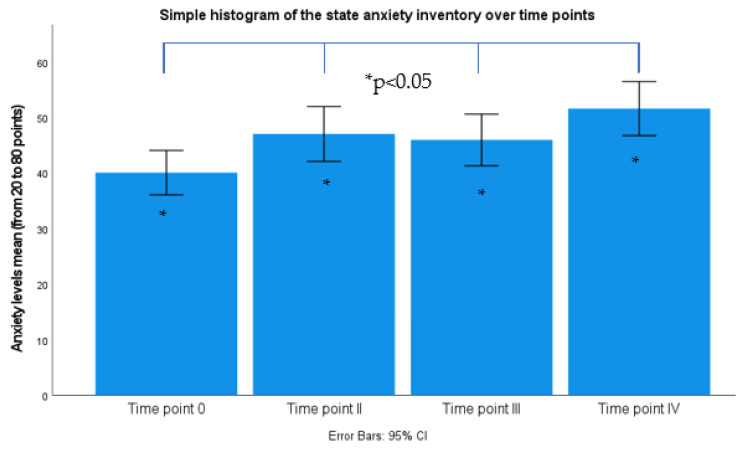
Simple Bar chart summaries of separate variables from anxiety level assessed with state anxiety inventory. Time point 0: 4 weeks prior exam; Time point II: before auriculotherapy; Time point III: 30 min after auriculotherapy and Time point IV: morning before examination, CI: confidence interval.

**Figure 5 healthcare-10-01816-f005:**
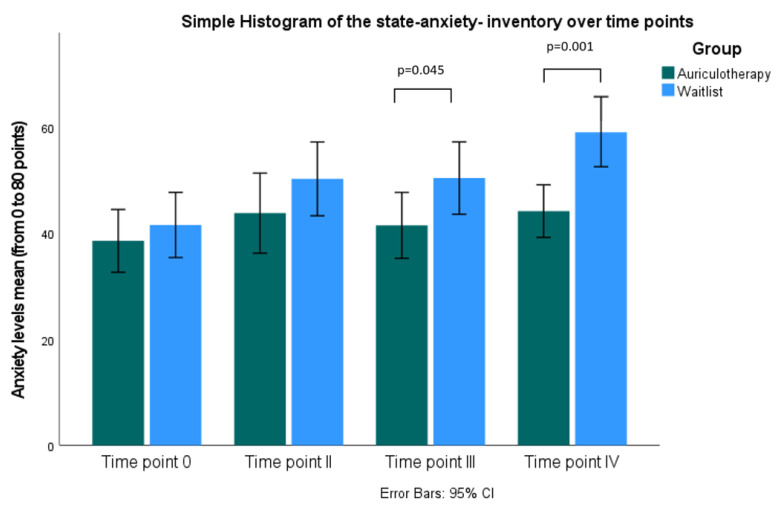
Histogram of separate anxiety levels measured using state anxiety inventory for both groups by independent samples *t*-test Time point 0: 4 weeks prior exam; Time point II: before auriculotherapy; Time point III: 30 min after auriculotherapy and Time point IV: morning before examination, CI: confidence interval.

**Figure 6 healthcare-10-01816-f006:**
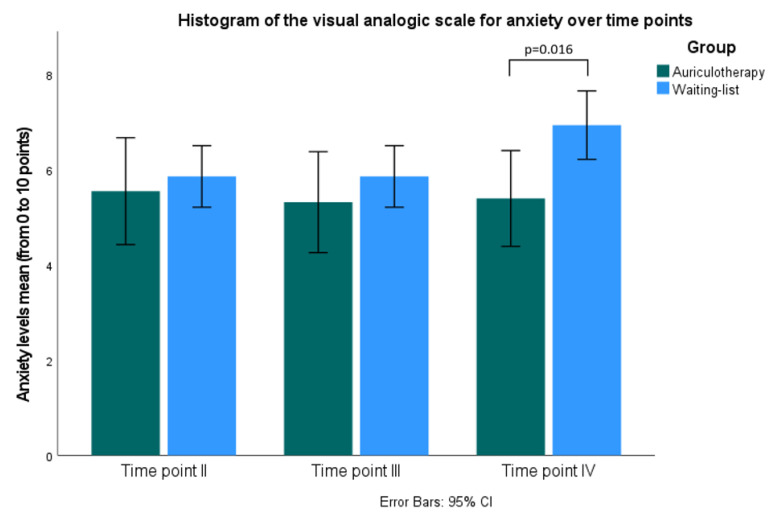
Histogram of separate anxiety levels measured using visual analogue scale for both groups by independent samples *t*-test. Time point II: before auriculotherapy; Time point III: 30 min after auriculotherapy and Time point IV: morning before examination, CI: confidence interval.

**Figure 7 healthcare-10-01816-f007:**
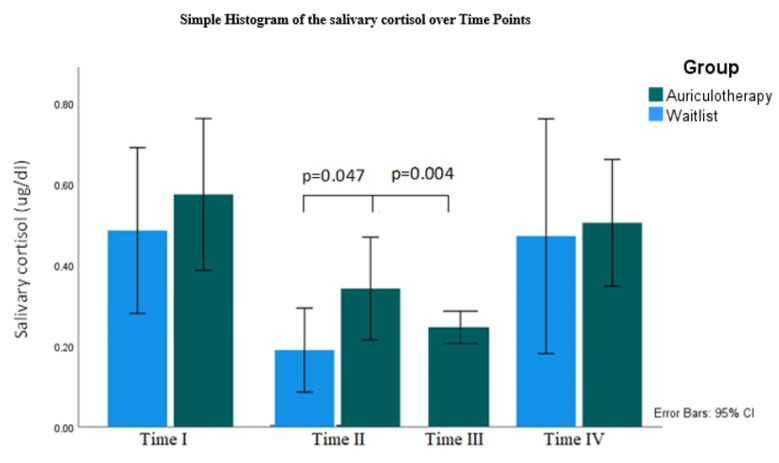
Mean of salivary cortisol by group, collected over four time periods, Time point I: morning before auriculotherapy; Time point II: before auriculotherapy; Time point III: 30 min after auriculotherapy and Time point IV: morning before examination, CI: confidence interval.

**Figure 8 healthcare-10-01816-f008:**
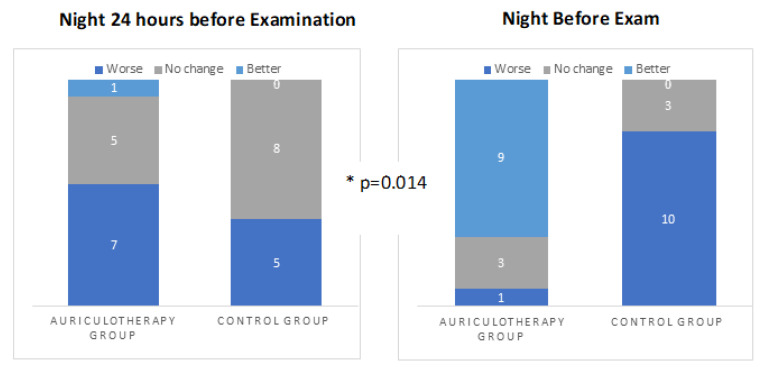
Simple bar chart summaries of separate variables by sleep quality.

**Table 1 healthcare-10-01816-t001:** Demographic data.

Variables	Categories	Auriculotherapy	Waiting List	Significance
Alcohol Intake	Yes No	0% 100%	0% 100%	
Coffee	*n* M (SD)	13 1.38 (1.26)	13 1.6 (1.3)	0.998
Smoking Habits	Yes No	66.7% 47.8%	33.3% 52.2%	
Height (cm)	*n* M (SD)	13 1.65 (0.08)	13 1.67 (0.76)	0.458
Weight (kg)	*n* M (SD)	11 68.18 (23.5)	13 67.15 (12.73)	0.892
Gender	Female Male	10 3	10 3	
Age	*n* M (SD)	12 20.08 (1.72)	12 20.38 (1.89)	0.683
STAI-Trace	M (SD) *n*	44.8 (7.9) 13	42.8 (8.4) 13	0.269 13

Footnote: *n* = participant’s number; M = mean; SD = Standardized mean.

**Table 2 healthcare-10-01816-t002:** Outcome of the investigation data by independent samples *t*-test given as mean ± standard deviations of anxiety levels measured by the Staite Anxiety Inventory Visual Analogue Scale.

Outcome	Time Points	Auriculotherapy Group M (SD)	Significance	Waiting List Group M (SD)
*n*		13		13
State Anxiety Inventory	I	38.5 (9.7)	0.451	41.5 (10.1)
	II	43.7 (12.4)	0.183	50.23 (11.5)
	III	41.4 (10.2)	0.046	50.3 (11.2)
	IV	44.1 (8.2)	0.001	59.0 (10.7)
Visual Analogue Scale	II	5.5 (1.85)	0.840	5.8 (1.1)
	III	5.3 (1.7)	0.479	5.8 (1.1)
	IV	5.3 (1.6)	0.016	6.9 (1.1)

Footnotes: *n* = participant’s number M = Mean; SD = Standard deviations; Time point 0 = 4 weeks before exam; Time point II = before auriculotherapy; Time point III = 30 min after auriculotherapy, and Time point IV = morning before examination.

**Table 3 healthcare-10-01816-t003:** Outcome of the investigation data by independent samples *t*-test given as mean and standard deviations of anxiety levels measured by salivary cortisol.

Time Points									
Outcome	Comparation	TP I	*p*	TPII	*p*	TP III	*p*	TP IV	*p*
Salivary cortisol from original data (ug/dL)	Auriculotherapy Group M (SD)	0.51 (0.28) *n* = 12	0.374	0.34 (0.17) *n* = 10	0.047	0.25 (0.06) *n* = 11	N.A	0.49 + 0.21 *n* = 12	0.431
	Waiting list group M (SD)	0.47 (0.20) *n* = 8		0.21 (0.12) *n* = 8		N.A		0.47 (0.31) *n* = 12	

Footnotes: *n* = participant’s number; M = Mean; SD = Standard deviations; Time point I = morning before auriculotherapy; Time point II = before auriculotherapy; Time point III = 30 min after auriculotherapy and Time point IV = morning before examination; *p* = significance level.

**Table 4 healthcare-10-01816-t004:** Outcome of the investigation data by pared samples *t*-test given as mean and standard deviations of anxiety levels measured by salivary cortisol.

Outcome	Comparation	*n*	M (SD)	*t*	*df*	*p*
Salivary cortisol from original data (ug/dL)	Par 1: TP I—TP IV (auriculotherapy group)	12	0.19 (0.31)	0.215	11	0.417
	Par 1: TP I—TP IV (waiting list group)	7	0.01 (0.18)	0.197	6	0.425
	Par 2: TP II—TP III (auriculotherapy group)	9	0.10 (0.13)	2.291	8	0.026

Footnotes: *n* = participant’s number; M = Mean; SD = Standard deviations; Time point I = morning before auriculotherapy; Time point II = before auriculotherapy; Time point III = 30 min after auriculotherapy, and Time point IV = morning before examination; *t =* distribution values for *t*-test; *df* = degrees of freedom *p* = significance level.

## Data Availability

Not applicable.

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
