# Peer review of "The Effect of Auriculotherapy on Situational Anxiety Trigged by Examinations: A Randomized Pilot Trial"

_healthcare, 2022, doi:10.3390/healthcare10101816_

Round 1

Reviewer 1 Report

The study design is well defined, and the statistical analysis is appropriate.
The review of Bibliography is satisfying enough.
The choice of stimulating two fixed points, either on the right and or on the left, based on the
reactivity of the site could be considered limitative in front of the wide range of active
auricular points. Nevertheless, the results confirm the effectiveness of auricular stimulation in
controlling anxiety, intended as a symptom.
Moreover, in the section of methods, it is not clearly described if the spheres, while in situ,
have been manually manipulated. These data would be important in better defining the
intensity of the effective stimulation.

In my opinion, the sample size is poor: 13 patients for arms are few in order to define the stastistical significance.
An increase of the sample size can give a higher level of evidence to the results.

Author Response

The team appreciate your relevant comments and feedback to improve our work. Please, find the suggested amendments accordingly:

 1) Methods section updated (line 136 and 137)
2) We also agree that the sample size is poor. This is a pilot study (described in the title and methodology) to provide evidence to perform the main study. Although, the pilot sample size was calculated based on the confidence level of 95%, margin of error 5%, population proportion of 50% and population size of 26. So, the minimum total sample size was 25 participants needed to have a confidence of 95% that the real value is within ± 5% of the measured value. We had 26 participants with statistical significance, proving that increasing the sample size would improve the results.

Thank you for your feedback,
Kind Regards
The main author

Reviewer 2 Report

Thank you for the opportunity to review this article.

This study is generally well performed and may have a clinical relevance, but the manuscript needs certain amendments. 

Introduction:

P2, paragraph 1. The explanation of the mechanisms should be stated more carefully and should cite more accurately the original articles, rather than relying on the reviews that cite them.

Generally, references should be cited more carefully. For example, ref. 26 provided inconclusive results and ref. 29 is a protocol of the study cited in ref. 30, etc.

Addressing anxiety levels as AL in the same sentences with AA (auricular acupuncture) may be confusing. 

Materials and Methods

P.4, Exclusion criteria: the word "No" from the first one, should probably be omitted. 

The authors mention there were 4 groups, stratifies according to STAI. Please mention the number of students in each group.

Secondary outcomes: 

If VAS for anxiety and STAI-Y1 are closely correlated, please explain why both of them were used.  

Saliva collection was described in minute details: if this is a new method of saliva collection, please state so, and if you followed an accepted protocol, please cite it instead of elaborating every step.

The assessment of sleep quality according to participant's impression how it was comparing to the previous week, may be not very reliable, as it is a subjective evaluation, and people tend to forget.

Results

Table 1 presents demographic data (and not the flow of participants, as referred in the text). It would be helpful if baseline STAI assessments could be included in this table to see whether there were differences between the groups.  

Fig. 4 is explained in the text, and thus may not be needed. 

Table 2 and Figs 5&6 provide the same information, and thus may be mutually exclusive. The same goes for Table 4 and Fig 7. 

Sleep quality: if the assessment may not be valid, and the results might reflect an expectancy of the treatment effect, it may be mentioned in the text, but possibly do not justify a separate Fig. 

Discussion, Limitations and Conclusions: 

These chapters should be more concise, addressing the results and the references more accurately

In Limitations the authors mention 30% of missing data from WG's salivary cortisol sample.  This is a large proportion of missing data, and should be a subject for statistical evaluation. 

1. Lines 213-216 The sample size assumption was twenty-six (26) participants per group. The results show thirteen (13) participants only. The difference would result in much lower statistical power of the tests. In such a case showing the actual statistical power results (Table 2) would be more appropriate. 2. Lines 217-229 The description includes a number of methods used in statistical analyses. The report of the results is expected therefore to be correlated to the methods used. It is recommended to either to rewrite this part of the methods, or to present the results accordingly. 3. Table 2 The results show a reduced stat power of four of the seven tests in both assessment methods: a) only TP I and TP IV in State Anxiety Inventory and TP IV in Visual Analog Scale require less than thirteen participants. Other tests would need much larger sample sizes to provide reliable results. 4. Line 424 The limitations and discussions should include the analyses of the above-mentioned discrepancies. The appropriate use of reliable results, as mentioned above, may have a positive impact on the possibility to perform a larger-scale trial in the future.

Author Response

The team appreciate your relevant comments and feedback to improve our work. Please, find the suggested amendments accordingly:

 Introduction:

P2, paragraph 1, line 46 - 50. The explanation of the mechanisms is cited more accurately.
Ref. 26 (inconclusive results) and ref. 29 (reference 30'protocol) were addressed accordingly.
(AL) was changed as anxiety levels throughout all manuscripts to avoid confusion.

Materials and Methods

P.4, Exclusion criteria: the word "No" was omitted. 

The number of students stratified in each group according to STAI - Trace were added.

Secondary outcomes: 

VAS for anxiety and STAI-Y1 are closely correlated, as referred in the manuscript; however, as two independent scales, and to find stability in the anxiety results, the authors have decided to use both outcomes, STAI as the primary outcome and VAS as secondary.  

Saliva collection followed an accepted protocol that was cited (line 181).

We agree that the assessment of sleep quality as it is a subjective evaluation. However, we decided to use sleep quality as Usichenko's (2020 - Auricular stimulation vs expressive writing for exam anxiety in medical students - A randomized crossover investigation) study also used (accepted published protocol). Moreover, as you can see on page 13, lines 366/367, we have suggested research on melatonin levels in future because of the subjectivity of the scale we used.

Results

Table 1 presents demographic data (not the flow of participants, as referred to in the text) – Changed (line 217). Baseline STAI – trace assessments included as requested.

"Fig. 4 is explained in the text and thus may not be needed." Thank you for your feedback, and the authors have decided to keep the legend in the figure.

"Table 2 and Figs 5&6 provide the same information and thus may be mutually exclusive. The same goes for Table 4 and Fig 7. Sleep quality: if the assessment may not be valid, and the results might reflect an expectancy of the treatment effect, it may be mentioned in the text, but possibly do not justify a separate Fig. "

We understand it can be excessive, but the authors decided to keep tables and figures as the figures will help the readers visualize, and the tables provide support for those authors in the future that might want to do a systematic review. 

Discussion, Limitations and Conclusions: 

Changes have been made throughout the manuscript as recommended. Could you please specify if anything else is needed? 

"In Limitations, the authors mention 30% of missing data from WG's salivary cortisol sample. This is a large proportion of missing data, and should be a subject for statistical evaluation."

 As referred to in statistical analyses, we have addressed the missing values >5%; they were inferred with multiple imputation analyses by the regression method. Please read lines 291 – 294, where we explained that the treated missed values had the same result (significative) as the original data; therefore, we used the original data with the missed values. We only reported the missed values to be transparent and honest so that subsequent researchers might consider this issue in the future.

"Lines 213-216 The sample size assumption was twenty-six (26) participants per group, and the results show thirteen (13) participants only. The difference would result in a much lower statistical power of the tests. In such a case showing the actual statistical power results (Table 2) would be more appropriate.

The statistical power depends on the power of the technique used, the magnitude differences, and "N". So, if the N presented has significant differences. Therefore, if we increase the "N", those statistical differences would be even higher statistical power. 
However, thank you for your comment; we realized by mistake we wrote 26 per group, but in fact, the pilot sample size was calculated based on the confidence level of 95%, margin of error 5%, population proportion of 50% and population size of 26. So, the minimum total sample size was 25 participants needed to have a confidence of 95% that the real value is within ± 5% of the measured value. We had 26 participants with statistical significance, proving that increasing the sample size would improve the results. Please, check lines 199 to 202 to see the amendment.

 2. Methods part was rewritten.

Points 3 and 4 addressed limitations and the conclusion section.

Thank you for your feedback,
Kind Regards
The main author